# Exploring the Relationship between Periodontitis, Anti-Periodontitis Therapy, and Extra-Oral Cancer Risk: Findings from a Nationwide Population-Based Study

**DOI:** 10.3390/biomedicines11071949

**Published:** 2023-07-10

**Authors:** Sung-Hsiung Chen, Jui-Feng Chen, Yu-Tung Hung, Tzu-Ju Hsu, Ching-Chih Chiu, Shu-Jui Kuo

**Affiliations:** 1Department of Orthopedic Surgery, College of Medicine, Chang Gung University, Kaohsiung Chang Gung Memorial Hospital, Kaohsiung 833401, Taiwan; chensh@cgmh.org.tw; 2Department of Education, China Medical University Hospital, Taichung 404327, Taiwan; dillon0501@gmail.com; 3Management Office for Health Data, China Medical University Hospital, Taichung 404327, Taiwan; yutunghung1227@gmail.com (Y.-T.H.); r7r0923.cmuh@gmail.com (T.-J.H.); 4School of Medicine, China Medical University, Taichung 404328, Taiwan; 5Department of Orthopedic Surgery, China Medical University Hospital, Taichung 404327, Taiwan

**Keywords:** periodontitis, periodontal therapy, prostate cancer, thyroid cancer, inflammation

## Abstract

This study aimed to evaluate the systemic impact of periodontitis, previously considered a local disease, on cancer occurrence. We enrolled 683,854 participants, comparing cancer incidence among those with and without periodontitis and assessing the impact of periodontal treatment on cancer risk. Regardless of gender, age, Charlson comorbidity index, or the use of non-steroidal anti-inflammatory drugs, periodontitis patients had a lower overall cancer risk than controls. However, men with periodontitis had a higher risk of prostate cancer (adjusted hazard ratio [aHR] = 1.22; 95% confidence interval [CI] = 1.10–1.35), and both men and women had a higher risk of thyroid cancer (women: aHR = 1.20, 95%CI = 1.04–1.38; men: aHR = 1.51, 95% CI = 1.15–1.99). Patients with periodontitis who received treatment showed a reduced cancer risk (aHR = 0.41; 95% CI = 0.38–0.44) compared to untreated patients. Proper treatment for periodontitis may lower an individual’s cancer risk more than if they did not have the disease at all, suggesting that periodontitis is a modifiable risk factor for cancer.

## 1. Introduction

Periodontitis is a progressive inflammatory oral disease caused by biofilm-forming microorganisms. It involves the gradual destruction of dental supporting tissues, including alveolar bone, periodontal ligament, cementum, and gingiva [1,2,3,4,5,6,7,8,9]. Periodontitis is the major cause of tooth loss in adults and the most prevalent form of bone pathology in humans [8]. Based on the Global Burden of Disease Study (2016), severe periodontal disease ranked as the 11th most common condition globally. The prevalence of periodontal disease was reported to range from 20% to 50% worldwide [10]. Gingivitis can progress to periodontitis in susceptible individuals [11]. Previously, periodontitis was regarded primarily as a localized disease confined to the oral cavity. However, recent literature has shed light on the link between periodontitis and systemic diseases, encompassing diabetes mellitus, cardiovascular disease, and various extra-oral remote cancers. This growing body of research has heightened interest in exploring the systemic effects of periodontitis outside the oral cavity [12,13,14].

The destructive inflammation in the disease process of periodontitis is driven by complement-dependent mechanisms following oral microbial dysbiosis and may translocate out of the oral cavity [1]. In periodontitis, bacterial plaque impairs the periodontal epithelium and allows the entry of oral pathogens and their harmful elements (endotoxins and exotoxins) into the bloodstream, thus inducing systemic inflammation [15,16]. Systemic inflammation has been identified as a crucial factor that facilitates the six well-established biological capabilities necessary for malignant transformation, including sustaining proliferative signaling, evading growth suppressors, resisting cell death, enabling replicative immortality, inducing angiogenesis, and activating invasion and metastasis [17]. Certain anti-inflammatory drugs have shown potential in preventing or reducing the risk of specific cancers in certain sites, such as colorectal, esophageal, gastric, biliary tract, and breast cancers [18]. These findings provide mechanistic clues into the connection between periodontitis, systemic inflammation, and the occurrence of cancer.

In our study, we aimed to compare the incidence of cancers remote to the oral cavity among subjects with or without periodontitis. We also tried to determine the impact of treatment for periodontitis, including subgingival curettage and periodontal flap surgery, on the occurrence of cancers. The rationale of this study was to investigate the impact of periodontitis on the occurrence of cancer remote to the oral cavity and to explore whether treatment for periodontitis has the potential to influence cancer occurrence among patients with periodontitis.

## 2. Materials and Methods

### 2.1. Data Source

This is a comparative cohort study utilizing data derived from the Longitudinal Generation Tracking Database (LGTD 2005), which is a subset of the Taiwanese National Health Insurance Database (NHIRD), consisting of 2 million individuals randomly selected from National Health Insurance (NHI) beneficiaries. The database contains their demographic characteristics, records of hospitalizations, outpatient visits, prescribed medications, treatment, and diagnosed diseases, and employs the International Classification of Diseases, 9th and 10th edition, Clinical Modification (ICD-9-CM and ICD-10-CM) as the coding system [19,20,21,22,23,24]. We also utilized the linked registries of the Registry for Catastrophic Illness Patient Database (RCIPD), which was also a subset of the Taiwanese NHIRD, to identify patients with cancer. This study was approved by the Institutional Review Board of China Medical University Hospital Research Ethics Committee (CMUH109-REC2-031(CR-2)).

### 2.2. Study Population

The study included a periodontitis cohort comprising patients newly diagnosed with periodontitis, indicated by disease codes ICD-9: 523.3 (aggressive acute periodontitis) and 523.4 (chronic periodontitis), or ICD-10: K05.2 (aggressive acute periodontitis) and K05.3 (chronic periodontitis), between 2000 and 2017. The aforementioned codes were assigned by dentists who submitted reimbursement claims to the NHI. These codes were determined based on the professional judgment and expertise of the dentists involved in the claims process. The diagnosis of periodontitis in our study was not based on the new classification of periodontal diseases, as the timeframe of our study spanned from 2000 to 2017. The comparison group consisted of individuals without a diagnosis of periodontitis. Propensity score matching was conducted in a 1:1 ratio, matching case groups to control groups based on sex, age (categorized at 5-year intervals), and the year of the index date. The index date was defined as the date when the aforementioned coding was first recorded for the newly diagnosed periodontitis patients. For the comparison group, the index date was set as the recruitment date after 1 January 2000, when the control subjects were enrolled for the analysis. The subjects who were under 20 years old or older than 100 years old and with missing gender and age data were excluded. All participants were prospectively followed from the index date of recruitment until their withdrawal from the NHI, occurrence of cancer, or 31 December 2017.

### 2.3. Main Outcome and Covariates

Different types of primary cancers were the main outcome in our research, including liver (ICD-9: 155, ICD-10: C22), breast (ICD-9: 174, ICD-10: C50.0, C50.1, C50.2, C50.3, C50.411, C50.412, C50.419), lung (ICD-9: 162, ICD-10: C33, C34, C7A.090), thyroid (ICD-9: 193, ICD-10: C73, E31.22), colon rectum (ICD-9: 153,154, ICD-10: C18, C19, C20, C21, C7A.02), prostate (ICD-9: 185, ICD-10: C61), kidney (ICD-9: 189, ICD-10: C64, C65, C66, C68, C7A.093), nasopharynx (ICD-9: 147, ICD-10: C11), stomach (ICD-9: 151, ICD-10: C16, C7A.092), bladder (ICD-9: 188, ICD-10: C67), skin cancer (ICD-9: 173, ICD-10: C44), and other cancers [25]. The aforementioned ICD-9 codings obtained from the LGTD 2005 dataset were validated through the linked registry RCIPD. Both the LGTD 2005 and RCPID databases originate from the mother NHIRD database.

Furthermore, we considered gender, age (20–39, 40–59, ≥60), Charlson comorbidity index (CCI) score, and non-steroidal anti-inflammatory drugs (NSAIDs) as confounding factors based upon the previous publication [25]. CCI quantifies comorbidity burden by assigning weights to conditions associated with increased mortality. It calculates a score by summing the weights for each comorbidity. CCI scores help assess health status, predict risk of mortality or complications, and adjust for comorbidity burden in research studies [26].

The impacts of periodontal surgeries for periodontitis, including subgingival curettage (scaling [claim code 91004C] and root planing [claim codes 91006C, 91007C, and 91008C]) and periodontal flap surgery (claim codes 91009B and 91010B), on the occurrences of cancer were also analyzed.

### 2.4. Statistical Analysis

The present study utilized a t-test to assess discrepancies in continuous variables and a chi-square test to evaluate variations in categorical variables between the periodontitis group and the control group. Power analysis was implemented to estimate the statistical power and significance level, with the latter being set at 0.001. The respective power values for sex, age group, CCI score, and NSAIDs were 0.760, 0.024, 0.999, and 0.954, respectively. The standardized mean difference (SMD), as determined by Cohen’s D, was employed to examine and compare the dissimilarities in variables between the two cohorts. The total sample’s standard deviation was used as the denominator in SMD calculations. Cohorts were considered to exhibit negligible differences if the SMD was below 0.1. Person-years (PYs) were defined as the cumulative sum of the individual years of observation experienced by each member within a specified population, and the incidence rate of cancers was computed by dividing the number of events by the PYs. Crude and adjusted hazard ratios (cHR and aHR) were calculated using single- and multi-variable Cox proportional hazard models, respectively. The Kaplan-Meier survival curve was utilized to compare cumulative incidence between the two cohorts, with differences assessed via the log-rank test. All statistical analyses were performed using SAS software (version 9.4), and plots were generated using R software (version 4.0) incorporating the “survival” package.

## 3. Results

We enrolled 683,854 participants in this study, with both cohorts comprising 341,927 subjects. The distributions of baseline characteristics, such as sex, age, CCI score, and exposure to NSAIDs, were comparable between the two groups (Table 1).

Males exhibited a higher hazard ratio for overall cancer compared to females. Individuals aged 40 to 59 and those above 60 showed higher cancer hazard ratios compared to those aged 20 to 39. Each additional year of age correlated with a 1.05-fold increase in overall cancer risk. Higher CCI scores and NSAID exposure were associated with an elevated risk of cancer. After adjusting for age, sex, CCI score, and NSAIDs, periodontitis patients had a significantly lower overall cancer risk (adjusted hazard ratio [aHR] = 0.66; 95% confidence interval [CI] = 0.64–0.67) compared to the comparison group (Table 2). Additionally, the cumulative incidence of overall cancer in the periodontitis group was notably lower than in the comparison group, as demonstrated by Figure 1 (*p* < 0.001 for log-rank test).

Table 3 presents the overall cancer risk in two groups, categorized by sex, age, CCI score, and exposure to NSAIDs. Regardless of sex, individuals with periodontitis had a lower cancer risk compared to the control group (female: aHR = 0.75, 95% CI = 0.73–0.77; male: aHR = 0.59, 95% CI = 0.57–0.60). This reduced risk was consistent across all age groups (20–39: aHR = 0.78, 95% CI = 0.73–0.82; 40–59: aHR = 0.66, 95% CI = 0.64–0.68; ≥60: aHR = 0.65, 95% CI = 0.63–0.68). Furthermore, the periodontitis cohort exhibited significantly lower cancer risk compared to the comparison group, regardless of CCI score (0: aHR = 0.66, 95% CI = 0.65–0.67; 1–2: aHR = 0.62, 95% CI = 0.57–0.67; ≥3: aHR = 0.77, 95% CI = 0.65–0.90) or NSAID use (non-NSAIDs: aHR = 0.67, 95% CI = 0.65–0.69; NSAIDs: aHR = 0.65, 95% CI = 0.63–0.67). In summary, patients with periodontitis demonstrated a lower risk of developing cancer overall, regardless of the stratification of sex, age, CCI score, or NSAID exposure.

Table 4 provides a detailed comparison of the risks of specific cancer types between individuals with periodontitis and the control group with sex-based stratification. Men with periodontitis exhibited a higher risk of prostate cancer (aHR = 1.22; 95% CI = 1.10–1.35). Furthermore, individuals with periodontitis, irrespective of sex, showed an increased risk of thyroid cancer (women: aHR = 1.20, 95% CI = 1.04–1.38; men: aHR = 1.51, 95% CI = 1.15–1.99). On the other hand, periodontitis patients, regardless of sex, had significantly lower risks of developing liver, lung, colon rectum, stomach, bladder, and other types of cancer when compared to those without periodontitis. Despite the overall trend of decreased cancer risk among periodontitis patients, there was an observed increase in the risk of specific cancer types, such as prostate cancer in males and thyroid cancer in both sexes.

Finally, we wanted to evaluate the influence of periodontitis treatment, including subgingival curettage (scaling [claim code 91004C]) and root planing (claim codes 91006C, 91007C, and 91008C), as well as periodontal flap surgery (claim codes 91009B and 91010B), on the overall occurrence of cancer in patients diagnosed with periodontitis [27]. After adjusting for age, sex, comorbidities, CCI score, and NSAID use, periodontitis patients who received treatment exhibited a lower cancer risk (aHR = 0.41; 95% CI = 0.38–0.44) compared to those without treatment, irrespective of age, CCI score, and NSAID usage (Table 5). The observed trend remained consistent across various stratifications, including sex, age, CCI score, and exposure to NSAIDs. Figure 2 shows that the cumulative incidence of cancer for periodontitis patients with treatment was significantly lower than those without (*p* < 0.001 for log-rank test). These findings suggest that periodontitis patients who undergo treatment have a significantly lower risk of developing cancer compared to those who do not receive treatment for periodontitis.

We aim to further explore the impact of periodontal treatment on the occurrence of prostate cancer and thyroid cancer in individuals with periodontitis. After adjusting for age, sex, comorbidities, CCI score, and NSAID use, periodontitis patients who underwent treatment demonstrated a reduced risk of prostate cancer (aHR = 0.60; 95% CI = 0.45–0.80) (Table 6) and thyroid cancer (aHR = 0.48; 95% CI = 0.28–0.80) (Table 7) compared to those who did not receive treatment.

Table 8 illustrates the overall cancer risk among patients with periodontitis, categorized based on treatment frequency. Compared to patients who did not receive any treatment, those who underwent treatment once exhibited an aHR of 0.46 (95% CI = 0.43–0.49), indicating a reduced cancer risk. Notably, patients who received two or more treatments demonstrated an even lower aHR of 0.28 (95% CI = 0.26–0.30). These findings suggest a clear association between treatment frequency and overall cancer risk among periodontitis patients, where higher treatment frequency corresponds to lower cancer risks.

## 4. Discussion

Our study found that individuals diagnosed with periodontitis have a lower risk of overall cancer than those without this diagnosis. Treatments for periodontitis, including subgingival curettage (scaling and root planing) and periodontal flap surgery, were found to mainly contribute to the risk reduction effects. Our study differs from previous research that suggests periodontitis is associated with a higher risk of overall cancer [18]. Our findings suggest that periodontitis may serve as a modifiable risk factor for cancer, and effective treatment could potentially reduce the risk compared to individuals without a periodontitis diagnosis. This study provides evidence supporting the notion that treating periodontitis extends beyond oral hygiene benefits and may have implications for cancer prevention. It is important to consider the public health implications of these findings, as interventions targeting the prevention and management of periodontitis could potentially contribute to reducing the risk of cancer development. Further research should explore the effectiveness of screening and treating periodontitis in reducing cancer risk. These findings offer a promising approach to cancer prevention and have potential implications for clinical practice and future research in this field.

Previous studies examining various ethnic groups have shown an overall trend of increased cancer risk among patients with periodontal disease compared to those without periodontitis [18]. The precise mechanism by which cancer may develop in individuals with untreated periodontitis remains unclear. Various potential mechanisms have been suggested, with systemic chronic inflammation appearing to play a prominent role in these proposed pathways. Periodontitis exemplifies an infectious process that can lead to systemic chronic inflammation if not properly addressed. Infections are recognized to trigger inflammation, and systemic chronic inflammation has been suggested as a mechanistic link between untreated periodontal disease and risk of cancer. Inflammatory processes could give rise to free radicals and active oxidative/nitrosative intermediates, which might contribute to DNA mutations within cells that interfere with DNA repair mechanisms [28]. The inflammatory cells themselves may further perpetuate the vicious cycle by producing free radicals, cytokines, chemokines, and metabolites of arachidonic acid, recruiting more inflammatory cells [28]. The oral microbiome and the oral-systemic link theories offer biological plausibility in terms of the connections between systemic chronic inflammation and cancer [18]. Specific periodontal pathogens like Fusobacterium nucleatum are associated with colorectal cancer, while Tannerella forsythia is linked to esophageal carcinoma [29,30]. In the oral-systemic link, ulcerated periodontal pocket walls can unintentionally allow toxic metabolites and oral bacteria to enter the bloodstream and affect distant body sites. Oral bacteria in the bloodstream, particularly their lipopolysaccharide component, can induce systemic inflammatory responses [31]. Inflammatory mediators released from periodontal disease, such as interleukin-6, tumor necrosis factor-alpha, and prostaglandin E2, can escape through damaged periodontal tissue pockets and produce systemic effects elsewhere in the body [32].

On the other hand, periodontal therapy has been shown to reduce plasma levels of interleukin-6, C-reactive protein, and fibrinogen in patients with severe periodontitis and refractory arterial hypertension [33]. Ryan et al. conducted a meta-analysis of randomized controlled trials to investigate the impact of anti-infective periodontal treatment on serum C-reactive protein levels. The results showed that periodontal treatment can lead to decreased final serum C-reactive protein levels compared to patients receiving no periodontal treatment. The effect was stronger in trials where the experimental group received antibiotics, with a mean difference of −0.75 mg/L (*p* = 0.03) [34]. The anti-inflammatory effects of periodontal therapy provide the mechanistic basis for the cancer risk-reducing effects of such therapy.

Based on our findings, we propose a hypothesis that suggests the presence of a basal level of systemic inflammation induced by the normal flora in the oral cavity of individuals without periodontitis. Conversely, untreated periodontitis is likely to significantly elevate systemic inflammation levels due to the presence of proliferated oral pathogens associated with the development of periodontitis. Nevertheless, rigorous periodontal therapy holds the potential to significantly diminish the abundance of oral microbes to a level even lower than that of the normal flora observed in individuals without periodontitis. As a result, periodontal therapy has the capacity to mitigate systemic inflammation to a degree below the baseline induced by the normal oral flora present in non-periodontitis individuals.

Our study has produced contrasting results compared to previous studies that did not account for the effects of periodontitis treatment. We believe the disparity between our findings and those of earlier studies can be attributed to variations in the intensity of periodontitis treatments received by patients. Treatment for periodontitis can significantly reduce the risk of cancer compared to individuals without periodontitis. Implementing standardized clinical measures to define cases of periodontal disease can be expensive and resource-intensive, especially in extensive surveillance studies. As a result, there is a lack of consistency in defining individuals with periodontal disease across epidemiological studies. Moreover, variations in study populations (such as ethnicity), sampling methods, study designs, and methodologies create difficulties in comparing findings across different studies and may contribute to the inconsistent results observed. We addressed the relevance of our findings in relation to the existing scientific evidence and the generalizability concerns in terms of case definition and ethnicity by providing a summary in Table 9.

Previous studies showed that treating periodontitis can reduce cancer risk in patients. In a study conducted by Huang et al., 38,902 periodontitis patients who received subgingival curettage (scaling and root planing) and periodontal flap surgery were selected. For each treatment cohort participant, two age- and sex-matched control cohort periodontitis patients were randomly selected. The authors found that the overall risk of developing cancer was significantly lower in the treatment cohort than in the untreated cohort (adjusted Hazard ratio = 0.72, 95% confidence interval = 0.68–0.76). The risks of developing most gastrointestinal tract, lung, gynecological, and brain malignancies were also significantly lower in the treatment cohort than in the comparison cohort [40]. However, the risks of prostate and thyroid cancers were significantly higher in the treatment cohort than in the comparison cohort. Huang’s study focused on treatment effects in periodontitis patients, whereas our study demonstrates that treated individuals with periodontitis may have a lower cancer risk compared to those without periodontitis. However, we observed higher risks of prostate and thyroid cancers in the periodontitis cohort, suggesting that the protective effects of anti-periodontitis treatment were less pronounced for these specific cancer types.

Periodontitis patients have a higher risk of thyroid cancer in both sexes and prostate cancer, suggesting that periodontal treatment does not provide a protective effect against these specific cancer types. Multiple dental X-ray exposures have been associated with an increased risk of thyroid cancer, according to recent studies [41]. Our hypothesis suggests that while periodontal treatment may reduce systemic inflammation, patients may still be exposed to dental X-rays, contributing to an overall elevated risk of thyroid cancer. In the case of prostate cancer, periodontal therapy may facilitate the dissemination of oral microbiota, including pathogens like Fusobacterium nucleatum, into the bloodstream, potentially infiltrating the prostate gland [42]. We propose that although periodontal treatment can alleviate systemic inflammation, the dissemination of oral pathogens to the prostate gland may predispose patients to prostate cancer.

The primary limitation of this study is our inability to obtain information on the history of smoking and alcohol use from the NHIRD. However, it is important to note that smoking is a major risk factor for periodontitis, and alcohol consumption is associated with moderately increased risk for periodontal disease [43,44,45]. It is highly plausible that individuals with periodontitis have a higher prevalence of smoking and alcohol consumption. It is unlikely that the potentially higher prevalence of smoking and alcohol consumption could account for the observed decreased risk of cancer among individuals with periodontitis. We anticipate that the observed association between periodontitis and a reduced risk of overall cancer would not be completely abrogated after adjusting for smoking and alcohol consumption. Generalizability is another important consideration. Our study was conducted on the Taiwanese population, which benefits from high accessibility to medical services and rigorous treatment for periodontitis. Therefore, it is crucial to exercise caution when extrapolating our findings to different ethnic groups or healthcare systems with varying levels of accessibility to periodontitis treatment compared to the Taiwanese system. Furthermore, it is plausible that the severity of periodontitis may differ between patients receiving treatment and those who are not, making the two groups incomparable in terms of homogeneity. However, we hypothesize that patients undergoing treatment are more likely to have a greater severity of periodontitis, accompanied by a greater extent of chronic systemic inflammation and an elevated risk of extra-oral cancer. The observed decrease in cancer risk within this treatment group can be seen as supportive evidence for the potential beneficial effects of periodontitis treatment on reducing extra-oral cancer risk. We acknowledge that confounding factors are ubiquitous challenges in epidemiologic association studies. Despite our efforts to adjust for various known confounders, it is possible that other unidentified factors could be influencing the observed outcomes. While we made extensive adjustments for a wide range of variables in our study, we cannot definitively claim to have accounted for every known or unknown factor that may influence the occurrence of cancer. This limitation is inherent to epidemiologic association studies.

## 5. Conclusions

We found that patients diagnosed with periodontitis exhibited a slightly lower overall risk of developing cancer, even after adjusting for age, sex, CCI score, and exposure to NSAIDs. An increased risk of overall cancer was observed among patients of advanced age and those with higher CCI scores. However, it is noteworthy that male patients with periodontitis had a higher risk of prostate cancer, and regardless of gender, there was an increased risk of thyroid cancer among periodontitis patients. Moreover, our study revealed that patients who received a greater number of treatments for periodontal disease experienced a significantly lower cancer risk compared to those who did not receive any treatment. These findings underscore the potential of periodontitis as a modifiable risk factor for cancer. Notably, our results suggest that appropriate treatment for periodontitis may not only mitigate an individual’s cancer risk but could potentially lower it to a greater extent than if they were free of the disease altogether.

## Figures and Tables

**Figure 1 biomedicines-11-01949-f001:**
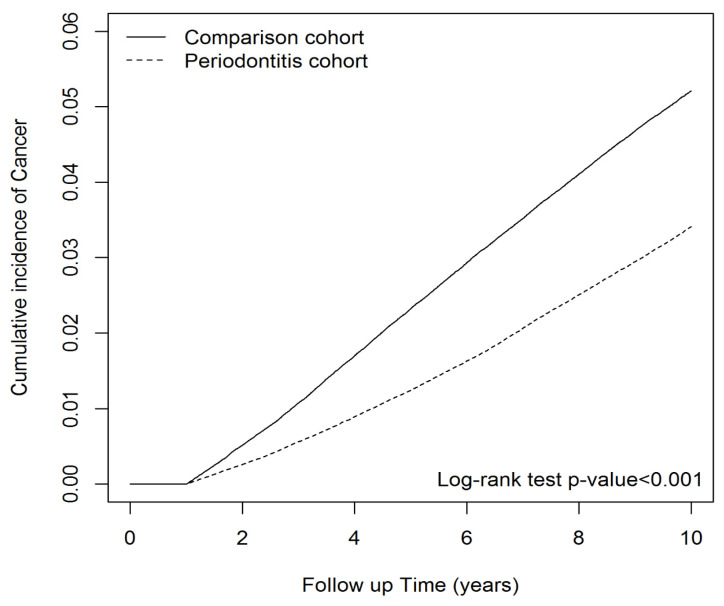
Cumulative incidence of cancer in periodontitis cohort and comparison cohort.

**Figure 2 biomedicines-11-01949-f002:**
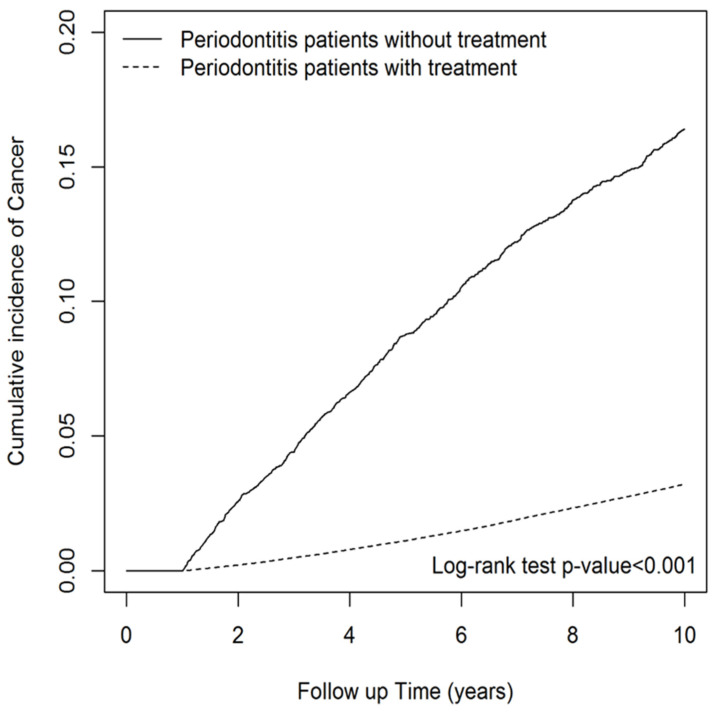
Cumulative incidence of overall cancer among the periodontitis patients stratified by the presence or absence of treatment.

**Table 1 biomedicines-11-01949-t001:** Baseline demographic characteristics of the individuals with and without periodontitis.

	Periodontitis				
	No (*N* = 341,927)		Yes (*N* = 341,927)		
Variables	*N*	%	*N*	%	SMD
Type					
Aggressive acute			178,243	52.13	
Chronic			163,684	47.87	
Sex					0.015
female	179,657	52.54	177,151	51.81	
male	162,270	47.46	164,776	48.19	
Age (years)					
20–39	144,681	42.31	146,258	42.77	0.009
40–59	144,368	42.22	143,577	41.99	0.005
≥60	52,878	15.46	52,092	15.23	0.006
Mean (SD)	43.49	(14.86)	43.36	(14.77)	0.009
CCI score					
0	321,297	93.97	326,267	95.42	0.065
1–2	15,982	4.67	13,285	3.89	0.039
≥3	4648	1.36	2375	0.69	0.066
NSAIDs					0.017
No	230,844	67.51	228,120	66.72	
Yes	111,083	32.49	113,807	33.28	
Follow-up duration: mean (SD)	11.46	(4.49)	12.16	(4.26)	0.160

CCI: Charlson comorbidity index; NSAIDs: non-steroidal anti-inflammatory drugs; SMD: standardized mean difference; SD: standard deviation.

**Table 2 biomedicines-11-01949-t002:** Hazard ratio of all cancers stratified by the history of periodontitis, sex, age, CCI score, and the exposure to NSAIDs.

Variables	All Cancers					aHR	95% CI	Mean Age
	N	PYs	IR	cHR	95% CI			
Periodontitis								
No	19,801	3,918,931.24	5.05	1.00	1	1.00	1	54.17
Yes	14,946	4,159,208.36	3.59	0.71	(0.69, 0.73) ***	0.66	(0.64, 0.67) ***	54.51
Type of periodontitis								
Aggressive acute	7792	2,166,313.6	3.60	1.00	1	1.00	1	55.14
Chronic	7154	1,992,894.8	3.59	1.00	(0.97, 1.03)	1.09	(1.05, 1.12) ***	53.81
Sex								
female	15,763	4,268,619.96	3.69	1.00	1	1.00	1	51.97
male	18,984	3,809,519.64	4.98	1.35	(1.32, 1.38) ***	1.30	(1.28, 1.33) ***	56.26
Age (years)								
20–39	4858	3,641,046.64	1.33	1.00	1	1.00	1	32.76
40–59	17,103	3,375,082.58	5.07	3.80	(3.68, 3.93) ***	3.77	(3.65, 3.90) ***	49.88
≥60	12,786	1,062,010.39	12.04	9.22	(8.92, 9.53) ***	8.47	(8.19, 8.76) ***	68.43
Mean (SD)				1.06	(1.05, 1.07) ***	1.05	(1.04, 1.06) ***	
CCI score								
0	31,803	7,774,788.18	4.09	1.00	1	1.00	1	53.64
1–2	2309	255,924.37	9.02	2.25	(2.16, 2.35) ***	1.05	(1.01, 1.09) *	61.31
≥3	635	47,427.06	13.39	3.53	(3.26, 3.82) ***	1.27	(1.18, 1.38) ***	62.76
NSAIDs								
No	16,746	5,438,338.71	3.08	1.00	1	1.00	1	54.44
Yes	18,001	2,639,800.89	6.82	2.22	(2.17, 2.27) ***	2.01	(1.97, 2.06) ***	54.19

CCI: Charlson comorbidity index; NSAIDs: non-steroidal anti-inflammatory drugs; PYs: person-years; IR: incidence rate per 1000 PYs; cHR: crude hazard ratio; aHR: adjusted hazard ratio; CI: confidence interval; SD: standard deviation (* *p* < 0.05 and *** *p* < 0.001).

**Table 3 biomedicines-11-01949-t003:** The incidence rate of all cancers among the subjects with and without periodontitis stratified by sex, age, CCI score, and the use of NSAIDs.

	All Cancers									
	Periodontitis (−)			Periodontitis (+)						
	N	PYs	IR	N	PYs	IR	cHR	95% CI	aHR	95% CI
Sex										
female	8735	2,111,218.85	4.14	7028	2,157,401.11	3.26	0.79	(0.76, 0.81) ***	0.75	(0.73, 0.77) ***
male	11,066	1,807,712.39	6.12	7918	2,001,807.25	3.96	0.64	(0.63, 0.66) ***	0.59	(0.57, 0.60) ***
Age (years)										
20–39	2671	1,791,697.09	1.49	2187	1,849,349.55	1.18	0.79	(0.75, 0.84) ***	0.78	(0.73, 0.82) ***
40–59	10,041	1,641,733.95	6.12	7062	1,733,348.62	4.07	0.67	(0.65, 0.69) ***	0.66	(0.64, 0.68) ***
≥60	7089	485,500.20	14.60	5697	576,510.18	9.88	0.66	(0.64, 0.69) ***	0.65	(0.63, 0.68) ***
CCI score										
0	17,978	3,760,147.35	4.78	13,825	4,014,640.83	3.44	0.72	(0.70, 0.74) ***	0.66	(0.65, 0.67) ***
1–2	1410	130,266.38	10.82	899	125,657.99	7.15	0.65	(0.60, 0.71) ***	0.62	(0.57, 0.67) ***
≥3	413	28,517.52	14.48	222	18,909.54	11.74	0.79	(0.67, 0.93) **	0.77	(0.65, 0.90) **
NSAIDs										
No	9608	2,670,301.91	3.60	7138	2,768,036.80	2.58	0.71	(0.69, 0.74) ***	0.67	(0.65, 0.69) ***
Yes	10,193	1,248,629.33	8.16	7808	1,391,171.56	5.61	0.69	(0.67, 0.71) ***	0.65	(0.63, 0.67) ***

CCI: Charlson comorbidity index; NSAIDs: non-steroidal anti-inflammatory drugs; PYs: person-years; IR: incidence rate per 1000 PYs; cHR: crude hazard ratio; aHR: adjusted hazard ratio; CI: confidence interval (** *p* < 0.01 and *** *p* < 0.001).

**Table 4 biomedicines-11-01949-t004:** The incidence rate of cancers in different locations among males and females.

	Periodontitis						Univariate			Multivariate		
	No			Yes								
Location	N	PYs	IR ^#^	N	PYs	IR ^#^	cHR	95% CI	*p*-Value	aHR ^$^	95% CI	*p*-Value
**Female**												
Liver	828	2,056,954.97	0.40	454	2,107,255.33	0.22	0.54	(0.48, 0.60)	<0.001	0.50	(0.44, 0.56)	<0.001
Breast (women only)	749	2,055,814.26	0.36	798	2,108,367.72	0.38	1.05	(0.95, 1.16)	0.364	1.01	(0.92, 1.12)	0.772
Lung	944	2,058,056.07	0.46	621	2,108,900.39	0.29	0.64	(0.58, 0.71)	<0.001	0.59	(0.54, 0.66)	<0.001
Thyroid	339	2,054,020.32	0.17	424	2,106,692.28	0.20	1.22	(1.06, 1.41)	0.006	1.20	(1.04, 1.38)	0.013
Colon rectum	1161	2,059,541.09	0.56	827	2,109,810.15	0.39	0.70	(0.64, 0.76)	<0.001	0.65	(0.59, 0.71)	<0.001
Kidney	206	2,053,053.23	0.10	199	2,105,224.60	0.09	0.94	(0.78, 1.15)	0.556	0.87	(0.71, 1.06)	0.155
Nasopharynx	82	2,052,138.94	0.04	97	2,104,381.47	0.05	1.16	(0.86, 1.55)	0.332	1.13	(0.85, 1.52)	0.401
Stomach	308	2,053,575.80	0.15	199	2,105,265.40	0.09	0.63	(0.53, 0.76)	<0.001	0.59	(0.50, 0.71)	<0.001
Bladder	149	2,052,705.58	0.07	117	2,104,632.39	0.06	0.77	(0.60, 0.98)	0.031	0.69	(0.54, 0.88)	0.002
Skin	92	2,052,269.36	0.05	110	2,104,568.93	0.05	1.17	(0.88, 1.54)	0.279	1.05	(0.80, 1.39)	0.718
Hematologic	467	3661.79	127.53	381	3403.48	111.94	0.81	(0.70, 0.92)	0.002	0.77	(0.67, 0.89)	<0.001
Other	3410	24,052.69	141.77	2801	22,045.13	127.06	0.83	(0.79, 0.87)	<0.001	0.84	(0.80, 0.88)	<0.001
**Male**												
Liver	2156	1,747,878.42	1.23	1076	1,948,148.38	0.55	0.45	(0.42, 0.48)	<0.001	0.42	(0.39, 0.45)	<0.001
Lung	1544	1,744,410.16	0.89	767	1,946,140.30	0.39	0.45	(0.41, 0.49)	<0.001	0.39	(0.35, 0.42)	<0.001
Thyroid	80	1,734,928.25	0.05	139	1,940,927.57	0.07	1.54	(1.17, 2.03)	0.002	1.51	(1.15, 1.99)	0.003
Colon rectum	1438	1,744,025.23	0.82	1232	1,949,306.76	0.63	0.77	(0.71, 0.83)	<0.001	0.68	(0.63, 0.74)	<0.001
Prostate (men only)	633	1,738,962.97	0.36	1060	1,947,929.48	0.54	1.49	(1.35, 1.64)	<0.001	1.22	(1.10, 1.35)	<0.001
Kidney	265	1,736,165.96	0.15	253	1,941,832.66	0.13	0.85	(0.72, 1.01)	0.066	0.75	(0.63, 0.89)	0.001
Nasopharynx	210	1,735,658.89	0.12	258	1,941,540.43	0.13	1.11	(0.92, 1.33)	0.268	1.07	(0.90, 1.29)	0.440
Stomach	509	1,737,658.91	0.29	355	1,942,672.45	0.18	0.62	(0.54, 0.71)	<0.001	0.54	(0.47, 0.62)	<0.001
Bladder	375	1,736,832.95	0.22	368	1,942,573.77	0.19	0.88	(0.76, 1.01)	0.077	0.77	(0.67, 0.89)	<0.001
Skin	122	1,735,250.78	0.07	123	1,940,834.89	0.06	0.89	(0.70, 1.15)	0.380	0.79	(0.61, 1.02)	0.067
Hematologic	600	4574.56	131.16	546	4799.06	113.77	0.77	(0.69, 0.87)	<0.001	0.76	(0.68, 0.86)	<0.001
Other	3134	20,676.16	151.58	1741	13,544.39	128.54	0.78	(0.73, 0.82)	<0.001	0.76	(0.71, 0.80)	<0.001

^#^: Per 1000 person-year. ^$^: Adjusted HR estimated by the model including the variables of sex, age, CCI score, and NSAIDs. Abbreviations: PY, person-years; IR, incidence rate; CI, confidence interval; HR, hazard ratio.

**Table 5 biomedicines-11-01949-t005:** Overall cancer in patients with periodontal disease with and without treatment, using the Cox proportional hazards model.

	All Cancer						Univariate			Multivariate		
	Treatment (−)			Treatment (+)								
	N	PYs	IR	N	PYs	IR	cHR	95% CI	*p*-Value	aHR	95% CI	*p*-Value
All	872	52,419.06	16.64	14,074	4,106,789.30	3.43	0.19	(0.18, 0.21)	<0.001	0.41	(0.38, 0.44)	<0.001
Sex												
Female	332	22,921.95	14.48	6696	2,134,479.16	3.14	0.20	(0.18, 0.23)	<0.001	0.42	(0.38, 0.48)	<0.001
Male	540	29,497.12	18.31	7378	1,972,310.13	3.74	0.19	(0.17, 0.21)	<0.001	0.40	(0.37, 0.44)	<0.001
Age												
20–39	65	7112.56	9.14	2122	1,842,236.99	1.15	0.11	(0.09, 0.14)	<0.001	0.11	(0.08, 0.14)	<0.001
40–59	351	20,637.00	17.01	6711	1,712,711.62	3.92	0.22	(0.19, 0.24)	<0.001	0.24	(0.21, 0.26)	<0.001
60+	456	24,669.51	18.48	5241	551,840.68	9.50	0.48	(0.44, 0.53)	<0.001	0.50	(0.45, 0.55)	<0.001
CCI score												
0	750	44,980.99	16.67	13,075	3,969,659.84	3.29	0.19	(0.17, 0.20)	<0.001	0.39	(0.36, 0.42)	<0.001
1–2	90	5688.46	15.82	809	119,969.52	6.74	0.40	(0.32, 0.50)	<0.001	0.51	(0.41, 0.64)	<0.001
3+	32	1749.62	18.29	190	17,159.93	11.07	0.57	(0.39, 0.83)	0.004	0.62	(0.42, 0.91)	0.014
NSAIDs												
No	397	31,247.32	12.71	6741	2,736,789.48	2.46	0.18	(0.16, 0.20)	<0.001	0.40	(0.36, 0.44)	<0.001
Yes	475	21,171.75	22.44	7333	1,369,999.82	5.35	0.22	(0.20, 0.24)	<0.001	0.42	(0.38, 0.46)	<0.001

**Table 6 biomedicines-11-01949-t006:** Prostate cancer in patients with periodontal disease with and without treatment.

Prostate Cancer											
Treatment (−)			Treatment (+)			Univariate			Multivariate		
N	PY	IR	N	PY	IR	cHR	95% CI	*p*-Value	aHR	95% CI	*p*-Value
58	285.85	202.91	1002	7816.65	128.19	0.42	(0.32, 0.55)	<0.001	0.60	(0.45, 0.80)	<0.001

**Table 7 biomedicines-11-01949-t007:** Thyroid cancer in patients with periodontal disease with and without treatment.

Thyroid Cancer												
	Treatment (−)			Treatment (+)			Univariate			Multivariate		
	N	PY	IR	N	PY	IR	cHR	95% CI	*p*-Value	aHR	95% CI	*p*-Value
All	15	63.73	235.35	548	4046.22	135.44	0.50	(0.30, 0.84)	0.008	0.48	(0.28, 0.80)	0.005
female	11	43.86	250.80	413	2965.51	139.27	0.47	(0.26, 0.86)	0.015	0.45	(0.24, 0.82)	0.009
male	4	19.87	201.27	135	1080.71	124.92	0.52	(0.19, 1.42)	0.201	0.59	(0.21, 1.66)	0.321

**Table 8 biomedicines-11-01949-t008:** The risk of overall cancer associated with the frequency of treatment of periodontal disease using the Cox proportional hazard model.

	All Cancer			Univariate			Multivariate		
Trx freq.	N	PYs	IR	cHR	95% CI	*p*-Value	aHR	95% CI	*p*-Value
0	872	52,419.1	16.64	1.00		-	1.00		
1	11,314	2,905,542.4	3.89	0.22	(0.20, 0.23)	<0.001	0.46	(0.43, 0.49)	<0.001
≥2	2760	1,201,246.9	2.30	0.13	(0.12, 0.14)	<0.001	0.28	(0.26, 0.30)	<0.001

Trx freq.: treament frequency; PYs: person years; IR: incidence rate; CI: confidence interval.

**Table 9 biomedicines-11-01949-t009:** Previous publications on the association between periodontitis and cancer occurrence with different case definition criteria and ethnicity.

	Design	Participants (Including Case Definition Criteria and Ethnicity Information)	Results
[35]	Prospective cohort	The study was prospectively performed between 1986 and 2004, with US male health professionals aged 40–75 years responding to questionnaires from the Harvard University School of Public Health, USA. Baseline and follow-up questionnaires were used to gather information on periodontal disease history, tooth loss, smoking status, dietary intake, and new cancer diagnoses.	In the main analysis, 48,375 men were included with a median follow-up of 17.7 years, excluding participants with pre-1986 cancer diagnoses (excluding non-melanoma skin cancer, n = 2076) and those with missing periodontal disease data (n = 1078). A total of 5720 cancer cases were documented, with the most common types being colorectal (n = 1043), skin melanoma (n = 698), lung (n = 678), bladder (n = 543), and advanced prostate (n = 541). Participants with periodontal disease had an increased risk of total cancer (aHR 1.14 [95% CI 1.07–1.22]). Significant associations were found for lung, kidney, pancreas, and hematological cancers in individuals with periodontal disease. Fewer teeth at baseline (0–16) increased the risk of lung cancer. In never-smokers, periodontal disease was associated with significant increases in total and hematological cancers but not lung cancer.
[36]	Prospective cohort	The study examined the relationship between baseline periodontal disease (measured by questionnaire-recorded tooth mobility) and incident cancers in 15,333 Swedish twins between 1963 and 2004. The authors employed co-twin analyses to account for familial factors and performed analyses specifically on monozygotic twins to address confounding by genetic factors.	The authors identified 4361 cancer cases over 548,913 person-years. Baseline periodontal disease was associated with increased risks of multiple cancers: 15% for total cancer (aHR = 1.15, 95% CI: 1.01, 1.32), 120% for uterine cancer (aHR = 2.20, 95% CI: 1.16, 4.18), 62% for colorectal cancer (aHR = 1.62, 95% CI: 1.13, 2.33), 106% for pancreatic cancer (aHR = 2.06, 95% CI: 1.14, 3.75), and 47% for prostate cancer (aHR = 1.47, 95% CI: 1.04, 2.07). Co-twin analyses revealed a 50% higher risk of total cancer in dizygotic twins with baseline periodontal disease (aHR = 1.50, 95% CI: 1.04, 2.17), whereas the association was weaker in monozygotic twins (aHR = 1.07, 95% CI: 0.63, 1.81). Similar patterns were observed for digestive tract cancers, suggesting the influence of genetic risk factors.
[37]	Case-control	The study utilized data from the Hospital-based Epidemiologic Research Program at Aichi Cancer Center, Japan, including 5240 cancer subjects and 10,480 age- and sex-matched noncancer controls. The cohort consisted of patients with 14 types of newly diagnosed cancer from 2000 to 2005, while the controls were new outpatients without cancer during the same period. Tooth loss was assessed via a self-administered questionnaire and categorized as ≥21 remaining teeth, 9 to 20 teeth, 1 to 8 teeth, and 0 teeth.	A decreased number of remaining teeth was associated with increased odds ratio (OR) of head and neck (OR, 1.68; 95% CI, 0.88–1.93; *p* = 0.055), esophageal (OR, 2.36; 95% CI, 1.17–4.75; *p* = 0.002), and lung (OR, 1.54; 95% CI, 1.05–2.27; *p* = 0.027) cancers.
[38]	Prospective cohort	This US cohort study recruited 65,869 women aged 54 to 86 years. Periodontal disease data were collected via self-report questionnaires from 1999 to 2003. Cancer outcomes were assessed until September 2013, with a maximum follow-up period of 15 years. The main outcomes were incident total cancers, and secondary outcomes included site-specific cancers that were adjudicated by physicians.	During an average 8.32-year follow-up, 7149 cancers were detected. History of periodontal disease was associated with increased risk of total cancer (aHR 1.14, 95% CI 1.08–1.20); similar findings were observed in the subgroup analysis of 34,097 never-smokers (aHR 1.12, 95% CI 1.04–1.22). Significant associations were found for breast (aHR 1.13, 95% CI 1.03–1.23), lung (aHR 1.31, 95% CI 1.14–1.51), esophagus (aHR 3.28, 95% CI 1.64–6.53), gallbladder (aHR 1.73, 95% CI 1.01–2.95), and melanoma skin (aHR 1.23, 95% CI 1.02–1.48) cancers.
[39]	Prospective cohort	In the US Health Professionals’ Follow-up Study, 19,933 never-smoking men self-reported periodontal disease and teeth number at baseline and follow-up. Cancer cases were identified during a 26-year follow-up period.	Men with baseline periodontitis experienced a 13% increased risk of total cancer, while those with advanced periodontitis (<17 remaining teeth) had a 45% increased risk. No associations were found with prostate cancer, colorectal cancer, or melanoma. A 33% higher risk of smoking-related cancers (lung, bladder, oropharyngeal, esophageal, kidney, stomach, and liver) was observed. Men with advanced periodontitis had a substantially higher risk of smoking-related cancers (HR = 2.57, 95% CI 1.56–4.21; *p* = 0.0002) compared to those without periodontitis and with ≥17 teeth. Advanced periodontitis was linked to increased risks of esophageal, head and neck, and bladder cancers (HR = 6.29, 95% CI 2.13–18.6).
	Prospective cohort	Participants were drawn from the US ARIC study, a prospective cohort of 15,792 individuals aged 44 to 66 years recruited from 1987 to 1989. This analysis included 7466 participants from the ARIC cohort who reported edentulism or underwent a dental examination during 1996–1998. Probing depth and gingival recession were measured at six tooth sites to assess periodontal disease severity. Incident cancers (n = 1648) and cancer deaths (n = 547) were identified during a median 14.7-year follow-up.	Severe periodontitis (>30% of sites with attachment loss >3 mm) was associated with an increased risk of total cancer (aHR = 1.24, 95% CI = 1.07–1.44). The association was particularly strong for lung cancer (aHR = 2.33, 95% CI = 1.51–3.60). Colorectal cancer risk was also elevated for severe periodontitis, especially among never smokers (aHR = 2.12, 95% CI = 1.00–4.47). Associations were generally weaker or not observed among black participants, except for lung and colorectal cancers, which showed similar associations by race. No associations were found for breast, prostate, or hematopoietic and lymphatic cancers.

## Data Availability

The data presented in this study are available on request from the corresponding author.

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
