# Peer review of "Exploring the Relationship between Periodontitis, Anti-Periodontitis Therapy, and Extra-Oral Cancer Risk: Findings from a Nationwide Population-Based Study"

_biomedicines, 2023, doi:10.3390/biomedicines11071949_

Round 1

Reviewer 1 Report

MAJOR ISSUES

1.       Line 95 (Materials & Methods – 2.4. Statistical analysis) – what do you mean by saying that „two-tailed tests were applied to assess the statistical significance“? Statistical singificance is set arbitrarily (usually at 0.05). If you mean by this that you applied some form of power analysis (compromise or criterion power analysis), then be specific. Furtermore, the sample size in this study is huge, measured in hundreds of thousands of participants (which is very good), but that also means that a standard level of significance at α = 0.05 just won't cut it. Therefore, you should compute required α given power, effect size and sample size. Once that is done, you should reinterpret the results accordingly.

2.       Lines 63-71 (Materials & Methods – 2.2. Study population) – you stated that the PO cohort consisted of patients who were newly diagnosed with periodontitis between 2000 and 2017. Four different codes were applied to designate the diagnosis of periodontitis. Please state if these codes correspond to the severity of the diseases. And if they do, please explain if there is any reason you didn't stratify the incidence rates of cancers and different types of cancers based on the severity of periodontal disease.  

3.       Line 173 (Discussion) – you say that your findings differ from findings reported in previous studies which suggest that periodontitis is associated with the increased risk of overall cancer. Please add relevant references to this statement.

4.       Lines 222-231 (Discussion, closing paragraph) – you say that the primary limitation of this study was your inability to obtain history of smoking and alcochol use from NHIRD, both of which are risk factors for periodontal disease. Both of these can also be considered as risk factors for cancer. I do not understand what you mean by saying that introducing these two factors would cause potential bias effects skewing results towards increased risk of cancer among periodontitis patients. Statistical bias is a term used to describe statistics that don't provide an accurate representation of the population. Some data is flawed because the sample it surveys doesn't accurately represent the population. The number of participants in this study is not a problem because you have extremely large sample (which is great). However, the sample still does not correctly represent the population because you were not able to collect some additional information, in this case about two major risk factors for periodontal disease and cancer. What I do not follow from your line of reasoning is based on what you hypothesize that adjusting for information on smoking and alcochol use would make the risk of cancer for periodontitis patients would be even lower than presented here?

5.       Lines 232-240 (Conclusions) – you concluded that periodontitis patients had significantly lower overall risk of cancer than non-periodontitis patients. You also said that the treatment of periodontitis might reduce the risk of cancer even more. So how do you explain that, while on one hand, the anti-inflammatory effects of periodontal therapy provide mechanistic basis for the cancer risk-reducing effects of such therapy, on the other hand the absence of disease (and thus the absence of inflammatory effects) may be related to increased risk of cancer? To me this is non-sequitur.   

MINOR ISSUES:

1.       Line 64 (Materials & Methods – 2.2. Study population) – please specify what does „PO“ (PO cohort) abbreviation stand for.  

2.       Line 65 (Materials & Methods – 2.2. Study population) – please state that the diagnosis of periodontitis was not done according to new classification of periodontal diseases because the follow-up period was between 200 and 2017.

3.       Line 98 (Results) – state the sample size like this „683,854“ instead of „683854“.

4.       Table 1 (Results) – please state which SD you used to compute SMD – was it SD of the total sample? Could you define SMD as "Cohen's d", "Hedges' g" or Glass's Δ?

5.       Table 3 & Table 4 (Results) – please specify in tables' legends what does *** stand for.

6.       Line 149 (Results) – use Simple Past Tense; instead of „Finally, we want to assess“ put „Finally, we wanted to assess...“

7.       Line 163 (Results) – use abbreviation; instead of „95% confidence interval“ put „95% CI“.  

8.       Lines 188-198 (Discussion – 3rd paragraph) – please refrain from stating numbers in discussion section, just comment the results. Providng adequate references should be enough for more interested readers.    

Minor editing of English language required.

Reviewer 2 Report

Chen et al describe a nation-wide study about the correlation between periodontitis and cance risk. in the Material and Methods they describe the database and their main outcomes. They describe that:  "Different types of primary cancers were the main outcome in our research, " followed by a summary of cancer types enden by ' and other cancers" . In their summary there is no mention of oral cancer, which would be, in my opinion be the first type of cancer to investigate since the chronic inflammation caused by periodontitis might be a risk factor for precancerous changes in the cells at the side of inflammation.

The "other cancer" group is almost half of the total group of patients with cancer (table 4). The authors should further elaborate on these types of cancer. Were skin cancers like squamous cell carcinoma and basal cell carcinoma also included?

Overall, they find that periodontitis patients have a lower incidence of cancer deveolpment. This even decreases with treatment of the periodontitis (fig 2). Is it  plausible that patients needing treatment of their periodontitis suffer from more severe periodontitis than patients who are not treated?

As the authors conclude in the discussion, the results in this study are in contrast to the generally accepted idea that patients with periodontits have a higher chance of cancer.

Since these are all association stdies from large databases, is it possible that other cofounding factors are in fact more important in their findings than periodontits in itself?

Reviewer 3 Report

Abstract: There is one point which can be confusing to the readers: the phrase "patients with periodontitis who received treatment had a lower risk of cancer than those without". Here, it's unclear what "those without" refers to. If it means "those without treatment," it would be good to make that explicit. If it refers to "those without periodontitis," it seems to contradict the earlier statement about periodontitis patients having lower overall cancer risk than controls (presumably, people without periodontitis). I would recommend clarifying this part to avoid any confusion.

Introductions section: A minor point for consideration is the flow and readability of the text. Some sentences could be simplified or broken down for easier reading and understanding. For example, the sentence "Gingivitis progresses to periodontitis via damage to alveolar bone and periodontal ligaments[3]." might be clearer as "Gingivitis can progress to periodontitis, which involves damage to the alveolar bone and periodontal ligaments[3]."

Materials and Method section: 1. The propensity score matching is mentioned, but a detailed explanation of how it was used in the study is not provided. Clarifying this could help understand the role it played in your analysis.

2. You have mentioned NSIADs as a confounding factor. I assume you meant NSAIDs (nonsteroidal anti-inflammatory drugs). Please verify and correct if necessary.

3. The reference to software used for statistical analysis and plotting (SAS and R, respectively) provides a good level of transparency. However, mentioning the specific packages or libraries used in R for the analysis might be helpful.

4. It would be beneficial to explain why those specific cancers were chosen for the study, as the rationale behind this selection is not immediately apparent.

5. Providing some context for the choice of covariates can be helpful for readers to understand why they were considered relevant.

Results section: 1. Please clarify what you mean by "non-inflammatory anti-inflammatory drugs" under NSAIDs. There might be a typographical error here.

2. While this section provides significant data, you could enhance the clarity of the results by incorporating more narrative explanation of the results. This can help guide the reader through the data and aid in understanding your findings.

Discussion section: 1. You could consider discussing the implications of your research findings in more depth. For instance, how might these findings impact current clinical practice or future research in this field? How could they potentially shape public health policies?

2. Although you have mentioned the potential link between periodontitis and cancer through systemic chronic inflammation, the mechanistic detail regarding the specific role of periodontitis in lowering the risk of most cancers but increasing the risk of prostate and thyroid cancers remains under-discussed. This aspect could be expanded upon if possible.

3. While you addressed the limitation of lacking data on smoking and alcohol use, other potential limitations could be discussed. For example, consider the study's generalizability - would the results be similar in different populations?

The language used in this paper is generally clear and easy to understand, indicating a good command of academic English. The paper adheres to scientific writing conventions, uses technical terminology correctly, and presents a logical flow of ideas.

However, there are a few areas where the language could be improved:

Syntax and Clarity: There are some lengthy sentences that could be broken down into shorter ones for better readability. For instance, the sentence "Previous studies have reported on the cancer risk reduction effects of treatment for periodontitis among periodontitis patients..." can be made clearer.

Consistency in terminology: It would be better to maintain consistency in the use of terminologies. For example, "non-inflammatory anti-inflammatory drugs" should be replaced by "non-steroidal anti-inflammatory drugs (NSAIDs)" which is the correct term.

Avoiding repetition: The phrase "individuals with periodontitis who receive adequate treatment may have a lower cancer risk than individuals without periodontitis" was repeated almost verbatim within two paragraphs. It's crucial to avoid such repetition to maintain reader engagement.

Punctuation: Attention should be paid to correct punctuation usage. For example, the sentence "If treated rigorously, the cancer risk could be even lower compared to individuals without a periodontitis diagnosis." should have a comma after "rigorously" to ensure clarity.

Clarify abbreviations: Make sure that abbreviations are clearly defined upon first use and used consistently thereafter. For instance, "CCI" is used several times before being defined as Charlson Comorbidity Index.

General proofreading: Some minor typos and grammatical errors are present that can be caught through careful proofreading. For example, "wothout" should be corrected to "without".

Overall, while the language used is of a good standard, these minor revisions will help to ensure the text is as clear and engaging as possible.

Round 2

Reviewer 1 Report

Dear authors,

I would like to thank you for addressing the issues I had raised and revising the manuscript. The manuscript is now improved, but I do believe that three major issues were not addressed properly. In my opinion, another round of revision is required. Below is the list of those major issues:

1.       The level of signficance at α = 0.05 is appropriate when we deal with such a large sample. Even if something comes out as statistically significant, the actuial effect might be very small or negligible. I would strongly recommend that you perform criterion power analysis to calculate adequate α given the categories listed in tables 1 and 2 and then additionally improve those tables. I expect that you will come with some different conclusions from those presented in this version of the manuscript.

2.       What is actually the probability of getting cancer for the participants diagnosed with acute aggressive periodontitis, chronic periodontitis or without periodontitis? You may insert that data in Results section (text or table legends, whichever you find more appropriate).

3.       Considering the major issue number 4 (sampling bias) from the first round of revision, it must be more clearly explained. Based on what you hypothesize that the observed asociation between periodontitis and a reduced risk of overall cancer could become even more pronounced after adjusting for smoking and alcochol consumption? I don't follow this

4.       Considering the majpr issue number 5 (conclusion) from the first round of revision, I think it still must be addressed properly. As I said before, while on one hand the anti-inflammatory effects of periodontal therapy provide mechanistic basis for the cancer risk-reducing effects of periodontal therapy, it is hard to follow why the absence of disease (and thus the absence of inflammatory effects) may be related to increased risk of cancer.

Author Response

Please see the attached file, thanks !

Reviewer 3 Report

Based on the excerpts retrieved from the revised manuscript, it appears that the authors have made several changes in response to the reviewers' comments. Here are some of the key points:

1.     Generalizability and Confounding Factors: The authors have added a discussion about the generalizability of their study and the potential influence of unidentified confounding factors on the observed outcomes.

2.     Main Outcome and Covariates: The authors have made changes to the section discussing the main outcome and covariates of their study.

3.     Risks of Specific Cancer Types: The authors have provided a detailed comparison of the risks of specific cancer types between individuals with periodontitis and the control group, with sex-based stratification.

The authors acknowledge that despite their efforts to adjust for various known confounders, it is possible that other unidentified factors could be influencing the observed outcomes. This is a common limitation in epidemiologic association studies, and it is good that the authors have acknowledged it. The manuscript indicates that patients diagnosed with periodontitis exhibited a significantly lower overall risk of developing cancer, even after adjusting for age, sex, CCI score, and exposure to NSAIDs. However, it is noteworthy that male patients with periodontitis had a higher risk of prostate cancer, and regardless of gender, there was an increased risk of thyroid cancer. This counterintuitive finding could be a point of concern and would benefit from further discussion and explanation.

However, this paper has merits. The authors have conducted a comprehensive analysis of the risks of specific cancer types in individuals with periodontitis compared to a control group. This detailed comparison, including sex-based stratification, provides valuable insights into the complex relationship between periodontitis and cancer risk. These merits make this paper a valuable contribution to the field of research exploring the link between periodontitis and cancer risk.

The authors' response to comments on the quality of English language used in the paper is appreciated. The revised manuscript would need to be evaluated for its readability and adherence to scientific writing conventions.

Author Response

Please see the attached file. Thanks !

Round 3

Reviewer 1 Report

I would like to thank the authors for improving the manuscript in the second round of revision. However, there are still some minor details that need to be addressed before the manuscript is ready for publishing.

1.       (Line 132-133) (2.4. Statistical analysis) – remove the sentence „A significance level of p<0.05 (two-tailed) was applied to determine statistical significance.“ The significance level of p<0.05 is no longer relevant.

2.       (Table 2) (3. Results, Table 2) – I do not understand what the „PYs column“ represents. Please clarify! Is it possible to list the mean age of participants for each category (periodontitis, sex, CCI score, NSAIDs).

3.       (Line 265-269) (4. Discussion) – Please remove any specific numbers or values from this paragraph in Discussion section (p-values, mean concentrations etc.). As long as the relevant reference is cited, more interested readers will be able to see additional information.

4.       (Line 336) (4. Discussion) – Instead of „completed abrogated“ it should be „completely abrogated“.

5.       (Line 357-359) (5. Conclusion) – instead of „a significantly lower overall risk of cancer“ I would put „a slightly lower risk of overall cancer“. Otherwise, the statement is a bit misleading. Based on IRs of all cancers presented in table 2, you should state the importance of age and CCI score for the increased risk of cancer.  

No comment.
